# Cytokinin Sensing in Bacteria

**DOI:** 10.3390/biom10020186

**Published:** 2020-01-25

**Authors:** Samar Kabbara, Baptiste Bidon, Jaafar Kilani, Marwan Osman, Monzer Hamze, Ann M. Stock, Nicolas Papon

**Affiliations:** 1Groupe d’Etude des Interactions Hôte-Pathogène (GEIHP, EA 3142), SFR ICAT 4208, UNIV Angers, UNIV Brest, F-49333 Angers, France; samarkabbara@hotmail.com (S.K.); baptiste.bidon@univ-angers.fr (B.B.); jaafar.kilani@univ-angers.fr (J.K.); 2Laboratoire Microbiologie Santé et Environnement (LMSE), Doctoral School of Science and Technology, Faculty of Public Health, Lebanese University, Tripoli 1300, Lebanon; marwan.osman@outlook.com (M.O.); mhamze@monzerhamze.com (M.H.); 3Department of Biochemistry and Molecular Biology, Center for Advanced Biotechnology and Medicine, Rutgers-Robert Wood Johnson Medical School, Piscataway, NJ 08854, USA; stock@cabm.rutgers.edu

**Keywords:** cytokinins, bacteria, two-component system, plant, biotic interactions

## Abstract

Although it has long been known that bacteria detect and react to plant chemicals to establish an interaction, the cellular signaling mechanisms involved in these perception processes have hitherto remained obscure. Some exciting recent advances in the field have described, for the first time, how some phytopathogenic bacteria sense the host plant hormones, cytokinins. These discoveries not only advance the understanding of cell signaling circuitries engaged in cytokinin sensing in non-plant organisms, but also increase our knowledge of the broad role of these ancient molecules in regulating intra- and interspecific communications.

## 1. Cytokinins Are Not only Plant Hormones!

Cytokinins (CKs) are well established as a prominent class of plant hormones (phytohormones) [1]. These adenine derivatives encompass more than forty different currently defined structures. CKs are known for their involvement in controlling many physiological processes in plants, influencing mainly growth and development such as regulation of cell division, control of morphogenesis/embryogenesis, and inhibition of senescence [2].

While the pleiotropic occurrence of CKs in plants is well documented, it is important to highlight that these molecules are also found in other organisms including bacteria, fungi, nematodes, insects, and even humans [1,3]. In these organisms, the roles and activities of CKs differ and knowledge of their functions remains fragmentary. For instance, there is increasing evidence of the roles of fungi-borne CKs in promoting fungal virulence [4]. Notably, this is observed in the biotic systems *Magnaporthe oryzae* (rice blast fungus)/*Oryza sativa* (rice) as well as in *Claviceps purpurea* (rye ergot)/*Secale cereale* (rye) [4,5]. Another interesting example of a pivotal role of CKs in biotic interactions is the ability of a plant-parasitic nematode to synthesize CK derivatives to manipulate the host system and establish long-term parasitic interactions [6,7]. Regarding the roles of CKs in amoebae, it was reported recently that the slime mold *Dictyostelium discoideum* produces six different CKs (notably *cis*-zeatin, isopentenyladenine, and discadenine) that coordinately orchestrate the different developmental stages of this protist [8,9,10]. However, studies of the occurrence and roles of CKs in non-plant organisms have been largely focused on bacteria [11]. With the exception of the human pathogen, *Mycobacterium tuberculosis*, which has been shown to produce CKs involved in virulence [12,13], most research has been conducted on phytopathogenic and plant symbiotic bacteria. Bacterial plant pathogens utilize two main types of CK production that result in profound metabolic and morphological plant modifications [3]. For instance, *Rhodococcus fascians* and *Rhizobium* spp. use a CK-mix production strategy, biosynthesizing many different CK derivatives to induce differentiated galls (known as “leafy galls”), and to establish symbiosis, respectively [14,15]. On the other hand, *Agrobacterium tumefaciens, Pseudomonas syringae, Ralstonia solanacearum*, and *Erwinia herbicola* employ a single CK production strategy during pathogenic processes [3].

## 2. Towards the Identification of a CK Signaling Pathway in Bacteria

Building upon the previous characterization of production and function of CKs in phytopathogenic and plant symbiotic bacteria, an exciting new development has been the pioneering description of how some phytopathogenic bacteria sense host plant CKs [16,17]. The gram-negative bacterium, *Xanthomonas campestris*, is the causative agent of black rot disease in crucifers (Figure 1A). It was presumed that the bacterium would sense plant-derived stimuli such as plant hormones for establishing infection, but until now, no perception mechanism had been revealed. A first and important step was taken in 2019 by Wang and colleagues, who described for the first time, a bacterial receptor (PcrK) that is capable of perceiving plant-borne isopentenyladenine in Xanthomonas (Figure 1B) [16]. PcrK belongs to the histidine kinase superfamily of proteins that act as primary sensors in a major class of bacterial regulatory circuitries referred to as “two-component systems” [18]. In the PcrK/PcrR regulatory system, binding of the plant hormone to the bacterial histidine kinase PcrK inactivates its autokinase activity, leading to dephosphorylation of the response regulator PcrR. The phosphodiesterase activity of PcrR (active when dephosphorylated) promotes the expression of many genes, including those involved in resistance to oxidative stress, thus enhancing bacterial resistance to host defenses (Figure 1B) [16].

One intriguing aspect of the PcrK protein is its apparent similarity to plant CK receptors, more specifically, similarity to the cyclase/histidine kinase associated sensor extracellular (CHASE) domain of the Arabidopsis AHK4 receptor [19]. Chen and colleagues established the CK recognition mechanism in Xanthomonas by determining the crystal structure of the CHASE domain of the PcrK receptor (Figure 1C) [17]. This allowed comparison to the CK-binding domain of the *Arabidopsis thaliana* homolog AHK4, the first structurally characterized plant CHASE domain [20]. Briefly, the study revealed that the Xanthomonas PcrK CHASE domain has a very similar topology to that of Arabidopsis AHK4 despite some differences, most notably the pocket size of the ligand binding site. Specifically, the ligand-binding pocket of the PcrK CHASE domain was captured in a “closed” conformation that prevents isopentenyladenine binding. The receptor presumably undergoes conformational changes in order to adopt an “open” state of the pocket (Figure 1C). These conformational changes are thought to be mediated by a periplasmic molecule which remains unidentified. Compared to that of Arabidopsis AHK4, the binding pocket of Xanthomonas PcrK is relatively small in size. This could explain why in vitro, PcrK is exclusively able to bind isopentenyladenine in contrast to Arabidopsis AHK4, which can also bind to a broad palette of CKs [17].

## 3. Concluding Remarks

While CKs have long been recognized as intra- and interspecific communication molecules, these compounds have primarily been associated with processes related to plant physiology [2]. This explains why historically, the first CK signaling pathway was deciphered in plants [19,21]. Although the syntheses and roles of CKs in non-plant organisms have been described over the past decades, the canonical CK signaling pathway, originally identified in Arabidopsis, was believed to be restricted to plants [1]. In this regard, the research studies recently published, concerning the characterization of the CK signal transduction circuitry in Xanthomonas, shed light on unprecedented perspectives in the field of microbiology by demonstrating that bacteria interacting with plants have developed similar signaling pathways for sensing plant host chemicals. CK sensing in Xanthomonas by a two-component system is certainly not an isolated case, as recently suggested by preliminary insights reported in cyanobacteria [22]. It is also highly likely that these hormonal signaling circuitries have evolved and diversified throughout the prokaryotic domain, as revealed by the observation of domain arrangements in CHASE domain-containing proteins in various prominent plant pathogenic or symbiotic bacteria (Figure 2) [23].

In conclusion, we are currently witnessing tremendous advances in the field of CK signaling in microorganisms. These recent discoveries must also be put in perspective of the explosion in the number of identified LONELY GUY genes in many different phyla, attesting to the extent of the action spectrum of CKs within the tree of life [24,25]. These data provide a foundation for investigations that will substantially increase our knowledge of the broad roles of these ancient molecules in regulating intra- and interspecific communications.

## Figures and Tables

**Figure 1 biomolecules-10-00186-f001:**
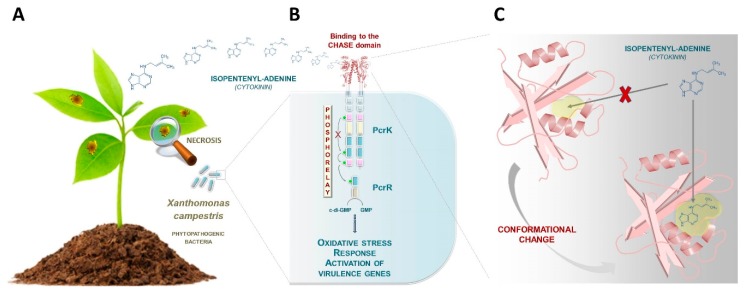
Towards the identification of a cytokinin (CK) signaling pathway in bacteria. (**A**) The gram-negative bacterium, *Xanthomonas campestris*, is the causative agent of black rot disease in crucifers. (**B**) Wang and colleagues have recently described a bacterial receptor (PcrK) that is capable of perceiving plant-borne isopentenyladenine in this plant pathogenic species [16]. When this plant hormone binds to the bacterial PcrK, the autokinase activity of PcrK is inactivated, leading to dephosphorylation of the response regulator PcrR. The phosphodiesterase activity of PcrR (active when dephosphorylated) leads to the expression of many genes, including those involved in the resistance to oxidative stress, thus enhancing bacterial resistance to host defenses. (**C**) Chen and colleagues determined the crystal structure of the cyclase/histidine kinase associated sensor extracellular (CHASE) domain of the PcrK receptor [17]. The ligand binding pocket of the PcrK CHASE domain was captured in a “closed” conformation that presumably undergoes a conformational change to reach the “open” state, which is capable of isopentenyladenine binding. These conformational changes are thought to be mediated by a periplasmic molecule which remains unidentified.

**Figure 2 biomolecules-10-00186-f002:**
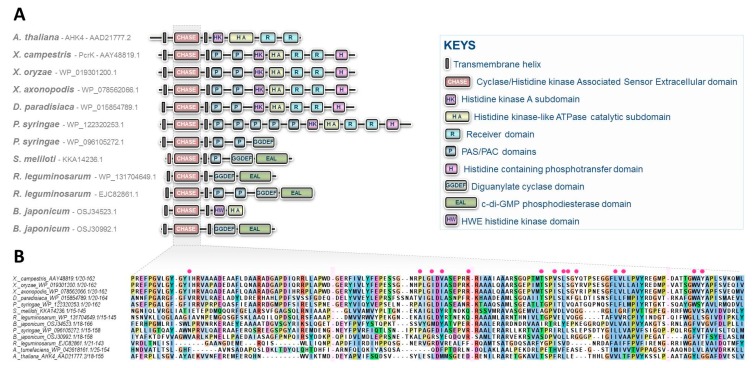
CHASE domain-containing proteins predicted in some prominent plant pathogenic or symbiotic bacteria. (**A**) Diversity of structures found in various plant interacting bacterial species including the recently characterized CK receptor, PcrK in Xanthomonas. The structure of the Arabidopsis CK receptor AHK4 is also provided for comparison. No CHASE domain-containing proteins are predicted in *Ralstonia solanacearum, Erwinia amylovora, Xylella fastidiosa, Pectobacterium* sp., *Rhodococcus fascians*, and *Spiroplasma* sp. (**B**) Alignment of CHASE domains from these predicted bacterial proteins. Interruptions of the alignment are indicated by pink rectangles, and essential residues reported to be involved in the binding pocket of Xanthomonas PcrK are indicated with red circles [17].

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
