# Peer review of "Cytokinin Sensing in Bacteria"

_biomolecules, 2020, doi:10.3390/biom10020186_

Round 1

Reviewer 1 Report

The commentary by Kabbara et al. titled “ Cytokinin Sensing in Bacteria” submitted to biomolecules is an interesting manuscript. I mentioned below my comments.

L12. At the begining of the abstract the fist letters is indicated in bold. Is it the journal style? If it is not, then change this typo.

L13-Please change the word “infection” by interaction.

L37 Please indicate as cis-zeatin, idnicating as cursive the position of the radical.

L82 Please indicate the year in which Wang discovered the bacterial receptor (PcrK)

L100 Xanthomonas is OK? Or the correct form in the manuscript should be in cursive letters?

L119 the subspecies (sp.) of the bacteria should be without cursive style? If so, please write it correctly.

- I think that a nice conclusion or perspectives can be added to the manuscript.

Author Response

REVIEWER 1 :

The commentary by Kabbara et al. titled “Cytokinin Sensing in Bacteria” submitted to biomolecules is an interesting manuscript. I mentioned below my comments.

L12. At the beginning of the abstract the first letters is indicated in bold. Is it the journal style? If it is not, then change this typo.

REPLY : We deleted the bold format in the abstract

L13-Please change the word “infection” by interaction.

REPLY : Correction was done

L37 Please indicate as cis-zeatin, indicating as cursive the position of the radical.

REPLY : We have italicized cis in cis-zeatin

L82 Please indicate the year in which Wang discovered the bacterial receptor (PcrK)

REPLY : We have indicated 2019

L100 Xanthomonas is OK? Or the correct form in the manuscript should be in cursive letters?

REPLY : In general, the sole genus name used to qualify an organism is not italicized.

L119 the subspecies (sp.) of the bacteria should be without cursive style? If so, please write it correctly.

REPLY : Correction was done.

- I think that a nice conclusion or perspectives can be added to the manuscript.

REPLY : We have already written a robust conclusion and perspectives in the last page, after figure 2. We believe that the reviewer did not see this part (?)

Reviewer 2 Report

This manuscript entitled “Cytokinin Sensing in Bacteria”, discuss the the cellular signaling mechanisms involved in how the bacteria detect and react to plant chemicals to establish an infection. The manuscript increase our understanding regarding cell signaling circuitries engaged in cytokinin sensing in non-plant organisms, as well as increase our knowledge of the role of these molecules play in regulating intra- and interspecific communications.

I think that the manuscript is suitable for publication in Biomolecules, as this journal focus on biogenic substances and their biological functions, structures, interactions with other molecules, and their microenvironment as well as biological systems. Also, the manuscript is well written and easy to follow.

Although the figures are very small and difficult to read. I also think that the figure should be better labelled. For example, Figure 1, right panel should rather be Figure 1C.

Author Response

REVIEWER 2 :

This manuscript entitled “Cytokinin Sensing in Bacteria”, discuss the cellular signaling mechanisms involved in how the bacteria detect and react to plant chemicals to establish an infection. The manuscript increase our understanding regarding cell signaling circuitries engaged in cytokinin sensing in non-plant organisms, as well as increase our knowledge of the role of these molecules play in regulating intra- and interspecific communications.

I think that the manuscript is suitable for publication in Biomolecules, as this journal focus on biogenic substances and their biological functions, structures, interactions with other molecules, and their microenvironment as well as biological systems. Also, the manuscript is well written and easy to follow.

Although the figures are very small and difficult to read. I also think that the figure should be better labelled. For example, Figure 1, right panel should rather be Figure 1C.

REPLY : We would like to thank the reviewer for his positive opinion on our manuscript. We have actually labelled subparts of figure 1 A, B, and C.

Reviewer 3 Report

Kaddara et all present a nice commentary on Cytokinins. They briefly review the history of CKs biology. They then nicely summary the findings by Wang et al. The commentary is well written and informative. The only thing I suggest to change is the resolution and the size of Fig 2- it is hard to read as it is.

Author Response

REVIEWER 3 :

Kabbara et al present a nice commentary on Cytokinins. They briefly review the history of CKs biology. They then nicely summary the findings by Wang et al. The commentary is well written and informative. The only thing I suggest to change is the resolution and the size of Fig 2- it is hard to read as it is.

REPLY : We would like to thank the reviewer for his positive opinion on our manuscript. We have actually increased the resolution of figure 2.